# Simple Sequence Repeats-Based Genetic Characterization and Varietal Identification of Potato Varieties Grown in Pakistan

**Aish Muhammad [1,\*], Saima Noor [1], Iqbal Hussain [1], Kazim Ali [1], Armaghan Shahzad [1], Mian Numan [1,2], Muhammad Zeshan [1], Syed Ijaz ul Hassan [3] and Ghulam Muhammad Ali [1]**

[1]  National Institute for Genomics & Advanced Biotechnology, National Agricultural Research Center, Islamabad 45500, Pakistan
[2]  College of Life Sciences, Shaanxi Normal University, Xi'an 710062, China
[3]  Potato Research Institute Sahiwal, Punjab 57000, Pakistan
\*  Correspondence: aishmuhammad@parc.gov.pk; Tel.: +92-51-90733807

**Abstract:** The potato (*Solanum tuberosum* L.) is an important cash crop with a complex genome and with features of aneuploidy with a high level of heterozygosity. It is a prerequisite for potato breeding to have knowledge of genetic diversity and population structure. To understand the genetic characteristics of potato cultivars in Pakistan, 25 potato varieties were characterized with simple sequence repeat (SSR) markers to distinguish closely related varieties. In total, 214 alleles were amplified with 35 SSR markers exhibiting 89.2% polymorphism. The maximum number of alleles and polymorphic alleles per locus were 20 and 14 for the markers S25 and S174, respectively. The polymorphic information content (PIC) extended from 0.00–0.87. The size of the amplified PCR product ranged between (30–1000 bp). A cluster analysis divided the 25 varieties into three clusters: cluster I revealed the most diversity, followed by cluster II with 11 varieties and cluster III with 13 varieties. Nei's genetic diversity and minimum spanning network (MSN) depicted the Mozika variety as the most diverse compared to the rest of the varieties. Nei's coefficient was found to vary from 0.53 to 0.95. Out of the 25 studied varieties, 16 were uniquely identified by 29 polymorphic SSR bands of different sizes with a maximum size amplified by S4026/4027 (800bp) and a minimum by S170 (90bp). The genetic diversity and varietal identification determined in the present study has molecular and breeding-related significance with respect to the utilization and protection of intellectual property rights of potato cultivars for sustainable potato production in Pakistan.

**Keywords:** potato; varieties; identification; genetic diversity; SSR marker; sustainability

## 1. Introduction

Potatoes (*Solanum tuberosum* L.) are among the leading commercial food crops globally after wheat, maize, and rice [1]. The increasing global population increases food demands [2], which is a serious challenge to world food security; thus, potatoes play a major role in coping with food security challenges. Potatoes are a major vegetable cash crop in Pakistan and are widely cultivated on an area of 0.31 million hectares with an annual production of 7.9 million tons [3]. More than 100 varieties are commercially cultivated in Pakistan, mainly from imported seed sources that are dependent on different countries. Potato varieties are evolving, producing higher yields, faster growth rates, and an enhanced pathogenic resistance compared to other crops. To date, most commercial potato varieties can be traced back to a limited number of popular cultivars, which represent a very narrow genetic base [4]. It is important for potato breeders to fully understand the genetic relationship among parental cultivars/lines and to choose elite parents of different genetic backgrounds for crossing in order to widen the genetic diversity of the potato population [5].

When increasing the number of potato varieties, it is important to distinguish them from each other. This identification is generally made based on traditional morphological

traits, such as the tuber shape, flesh color, skin, the depth of the eyes, flower color, sprouting, etc. Moreover, collecting such traits is laborious and time consuming, it is difficult to obtain accurate data, and trait collection is occasionally influenced by environmental factors [6]. Furthermore, biochemical markers such as isozymes are influenced by the growth conditions of a plant. Hence, the accurate identification of varieties is important for the development of new cultivars with inexpensive laboratory tools [7].

Therefore, the quick, detailed systematic identification and characterization of the cultivars are direly needed for safeguarding intellectual property rights, the cultivars' commercialization, and the production of seeds in Pakistan. There is a lack of accurate information and data concerning the molecular diversity and identification of potato varieties [8]. Several types of DNA markers are utilized for hereditary characterization and varietal identification, including Random Amplified Polymorphic DNA (RAPD) [9], Amplified Fragment Length Polymorphisms (AFLP) [10], Single Nucleotide Polymorphisms SNP [11], and Microsatellite Simple Sequence Repeats (SSR) [12].

SSR markers are the prevailing method for genetic applications, including diversity analysis, mapping, phylogenetic studies, cultivar difference, and Marker-Assisted Selection (MAS) in breeding programs [13]. SSR markers are very polymorphic and co-prevailing and the sequence of primers is preserved between and within the closely associated cultivars [6]. Moreover, the high heterozygosity and the ability to duplicate the ploidy status in potatoes are but a portion of the properties enabled by SSR markers [14]. Recently, specialists have dealt with SSR markers to save the genomic libraries and Expressed Sequence Tags' (ESTs) catalogs [6]. Furthermore, SSR markers are extensively utilized for DNA fingerprinting [15], parental investigation [16], genotype relocations [13], and genetic diversity analysis [12] in potatoes. According to the current status of research, many researchers are working on SSR markers' advancement and adaptability across potato varieties with respect to determining genetic diversity and varietal identification [17]

The goal of the current study on potatoes is to provide basic information for the breeding of genetic resources and conservation, to explore the genetic relationships and diversity among different potato cultivars, and to perform varietal identification using SSR markers. Although numerous studies have already been reported worldwide regarding potato DNA fingerprinting, no such data is available in Pakistan to date. The key objective of this research is to study the genetic relationship in the potato gene pool available in Pakistan, providing a systematized reference based on DNA fingerprints to establish unique genotypic identification. To assess the genetic diversity based on geography, we inferred the population structure analysis between the markers and the traits using a model-based approach indicating the distribution of alleles. The present study will be helpful for the management of the germplasm to improve its conservation, the design of cross combinations, genetic assessments, the production of elite cultivars and their registration, and varietal identification and its protection.

## 2. Materials and Methods

### 2.1. Sample Collection

Tubers from 25 different potato varieties—Mozika, Cardinal, Sahiwal White, PRI Red, Lady Rosetta, Kuroda, Rustum Red, Ronaldo, Desiree, Rock, Asterix, Ruby, Hermes, Sahiwal Red, Sante, Shepody, Rustum, Sadaf, Cosmos, Rustum white, LS, GN, J.8, Favorita, and Rocco—were collected from the Potato Research Institute, Sahiwal, and the Gansu Agricultural University, Lanzhou, China. The tubers were sown in a screenhouse at NIGAB (National Institute for Genomics and Advanced Biotechnology), Islamabad, Pakistan. The objective of this current study was to determine the diversity and perform the varietal identification of 25 local potato varieties grown in Pakistan.

### 2.2. DNA Extraction and SSR Analysis

Young leaves were collected after 4 weeks of germination for Genomic DNA extraction. DNA was extracted using 600 μL of 2% Cetyl-trimethyl Ammonium Bromide Buffer (CTAB)

(10 g/L CTAB salt (SERVA Electrophoresis GmbH D-69115 Heidelberg Carl-Benz-Str.7 Telefon 06221/138400) (20g/L of PVP (Phytotech Labs, 14610W, 106th St., Lenexa, KS, 66215, USA, Lot No. HYY0728019A), 0.5M/L of (PH 8) EDTA (Bio Basic Inc., Cat. No. 2616B024), 1 M/L (PH: 8) of Tris HCL (Invitrogen, 5781 Van Allen Way, Carlsbad, CA 92008, USA, REF, No. 15506-017), and 5 M/L of (PH 8) NaCl (Scharlab S.L. Gato Perez,33-P. I, Spain, Scharlau, cat no. 17137702) was dissolved in 1000 mL of distilled water PH 8 and then autoclaved. A total of 600 µL of Chloroform (Scharlab S.L. Gato Perez, 33-P. I, Spain, Scharlau, Cat. No. 02102500): Isoamyl alcohol (Bio Chemo pharma, ZA Conse Sur Ioire, 58200 France, cat. No. 209012500) (24:1) was added and the tubes were centrifuged at 13,000 rpm (Backman Coulter, Germany, cat. No. A46473) for 10 min at 15 °C. A total of 80 µL Sodium Acetate (Applichem, GmbH Ottoweg 4 D-64291 Darmstadt Germany, Lot. No. 0E003463) was added together with 520 µL of Chilled Isopropanol (Sigma Aldrich, St Louis MO 63178, USA, Cat. No. ZBJ7838) for rich precipitation of DNA. The DNA pellet was washed with 70% ethanol (Merck KGaA, 64271 Darmstadt, Germany, Lot. No. K47072383) and dissolved in Tris EDTA (Bio Basic Inc., Cat. No. 2616B024). DNA was quantified using 1% top vision agarose gel electrophoresis (Thermo Scientific, Baltics UAB VA Graciuno 8, LT-02241 Vihius, Lithuania, Cat. No. 01200415) with 2.5 µL of Ethidium Bromide (Thermo Scientific, Scharlau Chemie S.A. Gato Perez, 33-P-08181, Sentmenat Spain, European Union, CE Label, 10 mg/mL) for staining. The DNA bands were visualized by ultraviolet light through Gel Documentation System (Cleaver Scientific manufactured by Synoptics Ltd. Cambridge, UK). In the current study, 40 reported SSR markers synthesized from Macrogen company (Synbio Technologies LLC, 4250 US Route 1, Suite 3, Monmouth Junction, NJ 08852) were used (Supplementary Table S1).

*2.3. Polymerase Chain Reaction (PCR)*

A set of 35 reported SSR primers—as shown in (Supplementary Table S1)—were subjected to PCR amplification. PCR amplification was carried out with 50 µL Master Mix in 4.8 µL of (25 mM) $MgCl_2$ (Thermo Scientific, Baltics UAB VA Graciuno 8, LT-02241 Vihius, Lithuania), 5 µL PCR buffer $(NH_4)_2SO_4$ (10X) (Thermo Scientific, Baltics UAB VA Graciuno 8, LT-02241 Vihius, Lithuania), 0.6 µL of Taq DNA polymerase (5 U/µL) (Ferment Life Science, Baltics UAB VA Graciuno 8, LT-02241 Vihius, Lithuania Cat no. EP0402), 1.0 µL of dNTPs (25 mM) (Fermentas life Science, Hoffman-La Roche, U.S, Cat. No. 00056721), 4 µL (30 pmol/µL) of each Primer (Synbio Technologies LLC, 4250 US Route 1, Suite 3, Monmouth Junction, NJ 08852) and 4 µL of DNA template (50 ng/µL) from potatoes, and the total volume was maintained with 26.6 µL nuclease-free water (Invitrogen Thermo Fisher Scientific, 2130 Woodward St. Austin, TX 78744, USA, Cat. No. 2005355). Amplification was performed in 96-well plates (0.2 mL) (Thermo Scientific, Mexico, Cat. No. AB-0600). SSR-PCR amplification conditions were set accordingly followed by 35 cycles: the first stage involved denaturation at 94 °C for 1 min, the second stage was annealing at 52–56 °C for 45 s, and the third stage was extension at 72 °C for 45 s. The final extension was carried out at 72 °C for 10 min (Bio-Rad, Bio-Rad Laboratories, Inc., Singapore, Cat. No. 621BR62612). The PCR products were checked on a 3% agarose gel (Thermo Scientific, Baltics UAB VA Graciuno 8, LT-02241 Vihius, Lithuania Cat. No. 01200415) dissolved in 1x TBE buffer (10.9 g Tris Base (Invitrogen, Bio World, Genelinx International Inc, dba bioWORLD 4150 Tuller Rd. Suite 228 Dublin, OH 43017, Cat. No. 42020236-2), 5.5 g Boric Acid (Daejung, 186, Seohaean-ro, Siheung-si, Gyeonggi-do, Korea Lot. No. B0089PG2), and 0.745 g EDTA (Bio Basic Inc. company, U.S., Cat. No. 2616B024) dissolved in 1000 mL of distilled water with adjusted PH of 8.00) and the addition of 7.5 µL of ethidium bromide (Thermo Scientific, Scharlau Chemie S.A. Gato Perez, 33-P-08181, Sentmenat Spain, European Union, CE Label, 10 mg/mL) in agarose gel for staining, and confirmation products were visualized on 8% (PAGE) polyacrylamide gel for electrophoresis prepared by (30% solution were prepared by 29 g of Acrylamide (Roth, Carl Roth GmbH, Schoemperlenstr.3-5-D-76185 Karlsruhe, Cat. No. 7871), with 1g Bis acrylamide (Roth, Carl Roth GmbH, Schoemperlenstr.3-5-D-76185 Karlsruhe, Cat. No.

7867.2) and 10% APS (Roth, Carl Roth GmbH, Schoemperlenstr.3-5-D-76185 Karlsruhe) dissolved in 5x TBE buffer with 0.5% TEMED (Roth, Carl Roth GmbH, Schoemperlenstr.3-5-D-76185 Karlsruhe, Art. No. 2367.1) to investigate the DNA-banding patterns. The detected bands were visualized by ultraviolet light through Gel Documentation System (Cleaver Scientific manufactured by Synoptics Ltd. Cambridge, UK). The presence and absence of bands were counted via matching them with 50bp (Thermo Fisher Scientific, Baltics UAB VA Graciuno 8, LT-02241 Vihius, Lithuania Cat. No. EP0402) and 100bp plus DNA Ladder Plus (Thermo Fisher Scientific, Baltics UAB VA Graciuno 8, LT-02241 Vihius, Lithuania, Cat. No. SM0323) with well-known band sizes (50–15,000-bp).

### 2.4. Scoring and Data Analysis

A base file of 25 potato varieties based on SSR markers was made by assigning presence (1) and absence (0) of amplified products retrieved by the 35 SSR markers. The molecular size of each fragment was assessed with 50bp and 100bp DNA Ladder Plus (Invitrogen). Then, the genetic parameters were assessed, such as the percentage of the polymorphic locus (P) [18] and polymorphic information content (PIC) using formulas $p = (k/n) \times 100\%$, and $(PIC) = 1 - \sum i\, 1\, f\, 2i$, respectively [19]. A cluster diagram was constructed by an Un-Weighted Pair Group Method with Arithmetic Mean (UPGMA) and a Minimum Spanning Network (MSN) was formed by using POPPR software version 2.9.2 [20].

## 3. Results

### 3.1. Genetic Diversity Parameters Revealed Based on SSR Markers

The genetic characterization was achieved with 35 SSR primers that were selected from the international literature by considering the band size produced from various potato varieties. A total of 2473 scorable bands were amplified and 214 alleles of each locus ranging from 1 to 25 of each set of SSR primers were determined. The maximum number of scorable bands, 127, was detected from the S25 primer, S168 primer (124) allele, S182 marker (123), S187 (116), S118 (115), S174 (112), and S188 (100). The minimum number of alleles per locus was detected from STM1104 (21) and STM0031 (22). The minimum (0.33) allele frequency at each locus was obtained from the STI0032 marker and the maximum 100 allele frequency at each locus was revealed from the STM1104 primers. The genetic diversity ranged from 0.537–0.95, which was observed between the 25 varieties of potato, and the average was 0.413. The lowest PIC (0.00) value was obtained with the S180 marker, and the highest PIC (0.87) value was obtained with marker 4026/4027. A total of 16 potato varieties out of the 25 analyzed revealed unique alleles. An 89.22% rate of polymorphism was found in the 25 varieties of potato locally grown in Pakistan. (Table 1).

**Table 1.** Feature of SSR primers used for different 25 potato varieties.

| Primer | Annealing Temp (°C) | Total No. of Scorable Bands | Total no. of Alleles N | No. of Monomorphic Alleles | No. of Poly-Morphic Alleles = K | Ratio of Polymorphic Loci P | Size bp | Unique Bands | PIC |
|---|---|---|---|---|---|---|---|---|---|
| **S122** | 52 | 62 | 4 | 2 | 2 | 50 | 130–190 | 0 | 0.65 |
| **S168** | 52 | 124 | 8 | 2 | 6 | 75 | 50–400 | 0 | 0.83 |
| **S151** | 52 | 102 | 6 | 1 | 5 | 83.33 | 50–300 | 0 | 0.80 |
| **S184** | 52 | 98 | 8 | 0 | 8 | 100 | 170–550 | 1 | 0.81 |
| **S118** | 52 | 115 | 5 | 2 | 3 | 60 | 40–260 | 0 | 0.79 |
| **S192** | 52 | 91 | 8 | 0 | 8 | 100 | 180–310 | 1 | 0.83 |
| **S174** | 56 | 103 | 14 | 1 | 13 | 92.85 | 120–310 | 2 | 0.86 |
| **S25** | 56 | 127 | 20 | 1 | 19 | 95 | 160–800 | 8 | 0.87 |
| **S182** | 52 | 123 | 11 | 0 | 11 | 100 | 130–370 | 0 | 0.87 |
| **S187** | 52 | 116 | 8 | 2 | 6 | 75 | 40–350 | 1 | 0.83 |
| **S162** | 52 | 44 | 3 | 0 | 3 | 100 | 160–390 | 0 | 0.65 |
| **S148** | 52 | 69 | 9 | 0 | 9 | 100 | 180–490 | 1 | 0.86 |
| **S188** | 52 | 100 | 11 | 0 | 11 | 100 | 90–400 | 2 | 0.86 |

**Table 1.** *Cont.*

| Primer | Annealing Temp (°C) | Total No. of Scorable Bands | Total no. of Alleles N | No. of Monomorphic Alleles | No. of Poly-Morphic Alleles = K | Ratio of Polymorphic Loci P | Size bp | Unique Bands | PIC |
|---|---|---|---|---|---|---|---|---|---|
| S4026/4027 | 52 | 80 | 12 | 0 | 12 | 100 | 340–1000 | 3 | 0.87 |
| S8242 | 52 | 71 | 9 | 0 | 9 | 100 | 200–490 | 3 | 0.76 |
| S12002 | 56 | 68 | 4 | 1 | 3 | 75 | 40–245 | 0 | 0.68 |
| 31924 | 56 | 67 | 5 | 0 | 5 | 100 | 150–260 | 0 | 0.74 |
| 43016 | 54 | 49 | 5 | 0 | 5 | 100 | 50–210 | 1 | 0.73 |
| 46514 | 54 | 65 | 4 | 0 | 4 | 100 | 130–220 | 0 | 0.68 |
| STM0037 | 52 | 36 | 2 | 0 | 2 | 100 | 70–90 | 0 | 0.47 |
| STM0031 | 52 | 71 | 8 | 0 | 8 | 100 | 50–300 | 2 | 0.79 |
| ST10032 | 52 | 55 | 4 | 0 | 4 | 100 | 50–140 | 1 | 0.65 |
| S170 | 52 | 68 | 5 | 0 | 5 | 100 | 70–150 | 1 | 0.74 |
| S189 | 56 | 50 | 4 | 0 | 4 | 100 | 190–220 | 1 | 0.66 |
| S183 | 50 | 48 | 3 | 0 | 3 | 100 | 170–190 | 0 | 0.62 |
| S120 | 52 | 55 | 3 | 2 | 1 | 33.33 | 50–200 | 0 | 0.59 |
| S153 | 52 | 43 | 3 | 0 | 3 | 100 | 150–190 | 0 | 0.62 |
| 164010 | 56 | 49 | 3 | 2 | 1 | 33.33 | 30–250 | 0 | 0.57 |
| STM1106 | 52 | 82 | 6 | 1 | 5 | 83.33 | 40–200 | 0 | 0.78 |
| STM5114 | 52 | 67 | 3 | 1 | 2 | 66.66 | 40–310 | 0 | 0.66 |
| S7 | 52 | 71 | 6 | 0 | 6 | 100 | 50–190 | 0 | 0.79 |
| S180 | 54 | 22 | 1 | 0 | 1 | 100 | 190 | 0 | 0 |
| STM1104 | 54 | 21 | 2 | 0 | 2 | 100 | 180–190 | 0 | 0.36 |
| STM1052 | 52 | 28 | 4 | 0 | 4 | 100 | 210–250 | 1 | 0.36 |
| ST10004 | 52 | 33 | 3 | 0 | 3 | 100 | 90–110 | 0 | 0.64 |
| Total | | 2473 | 214 | 18 | 196 | 3122.83 | | | 24.4 |
| Average | | 70.65714 | 6.114286 | 0.514286 | 5.6 | 89.2237 | | | 0.69 |

The total of 35 SSR markers detected a total number of alleles = 2473, and the Total number of polymorphisms of the 35 SSR markers = 89.23.

### 3.2. Cluster Analysis

A cluster analysis was performed based on the data's similarity; the results are shown in the Dendrogram (Figure 1). A combination of the allele frequency loci was used to investigate the Genetic distance for the 25 varieties of potato. The cluster diagram was constructed by the unweighted pair group method with the arithmetic mean (UPGMA) and a minimum spanning network (MSN) was created using the POPPR software version 2.9.2 (Figure 2). Nei's coefficient was found to vary from 0.537 to 0.950 (Figure 2). The cluster analysis divided the 25 genotypes into three clusters: cluster I observed the most diversity, followed by cluster II containing 11 genotypes and cluster III containing the largest 13 genotypes, respectively. Nei's genetic diversity and minimum spanning tree determined that Mozika was the most diverse with respect to the rest of the genotypes. The variation patterns explored among potato varieties form a useful tool for distinguishing one potato variety from among others. Cluster 1 contains cv. Mozika while Cluster II was divided into two sub-clusters. The first one combined a variety of Ruby and Rustum White, whereas the second was subdivided into three groups. The first subgroup comprised the varieties Favorita and Rocco; the second included the J.8, LS, and GN; and the third group contained Kuroda, Rustum Red, Desiree, and Asterix varieties. Cluster III was further divided into four sub-groups. The first comprised Ronaldo, Lady Rosetta, and Hermes varieties; the second group contained Cardinal, Sahiwal White, and PRI Red varieties; the third group contained Sadaf, Shepody, and Rustum varieties; and the fourth group comprised four varieties, namely, Cosmos, Sante, Rocco, and Sahiwal Red. However, the Mozika variety was the most diverged compared to all other potato varieties. The maximum genetic distance (0.950) was calculated between Mozika and Sahiwal Red. A higher genetic similarity was

also shown in the Mozika and Ruby varieties at the molecular level (Figure 1). Therefore, it could be a useful variety due to its better potential for adaptability against the changing environment.

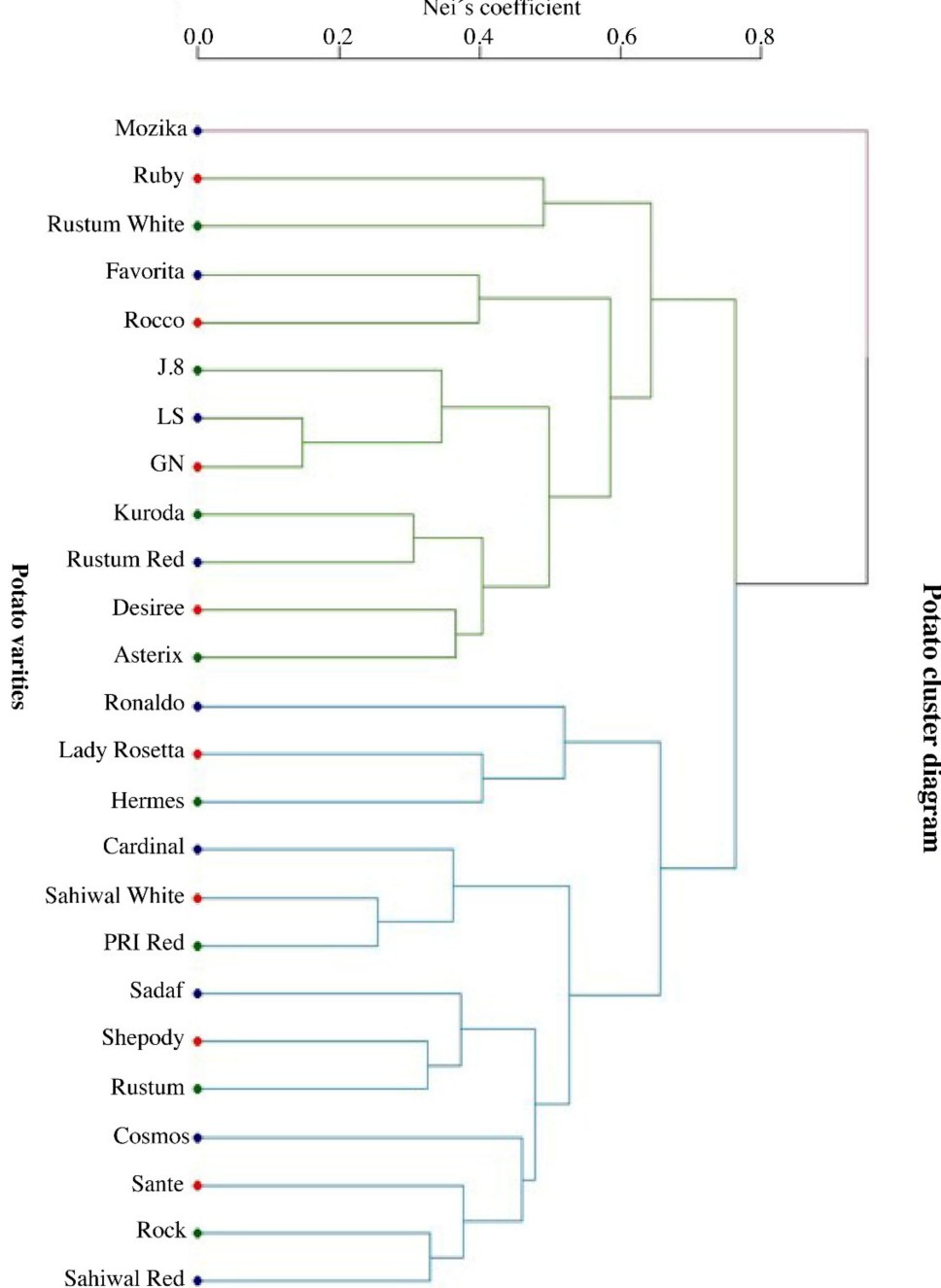

**Figure 1.** Diagrammatic representation of cluster dendrogram based on Nei's coefficient with 25 tips and 24 internal nodes. Nei's coefficient was found to vary from 0.537 to 0.950. Cluster analysis divided the 25 genotypes into three clusters. The Mozika variety is the most diverged compared to all other potato varieties. The maximum genetic distance (0.950) was calculated between Mozika and Sahiwal Red. A higher genetic similarity was also shown in Mozika and Ruby varieties at the molecular level.

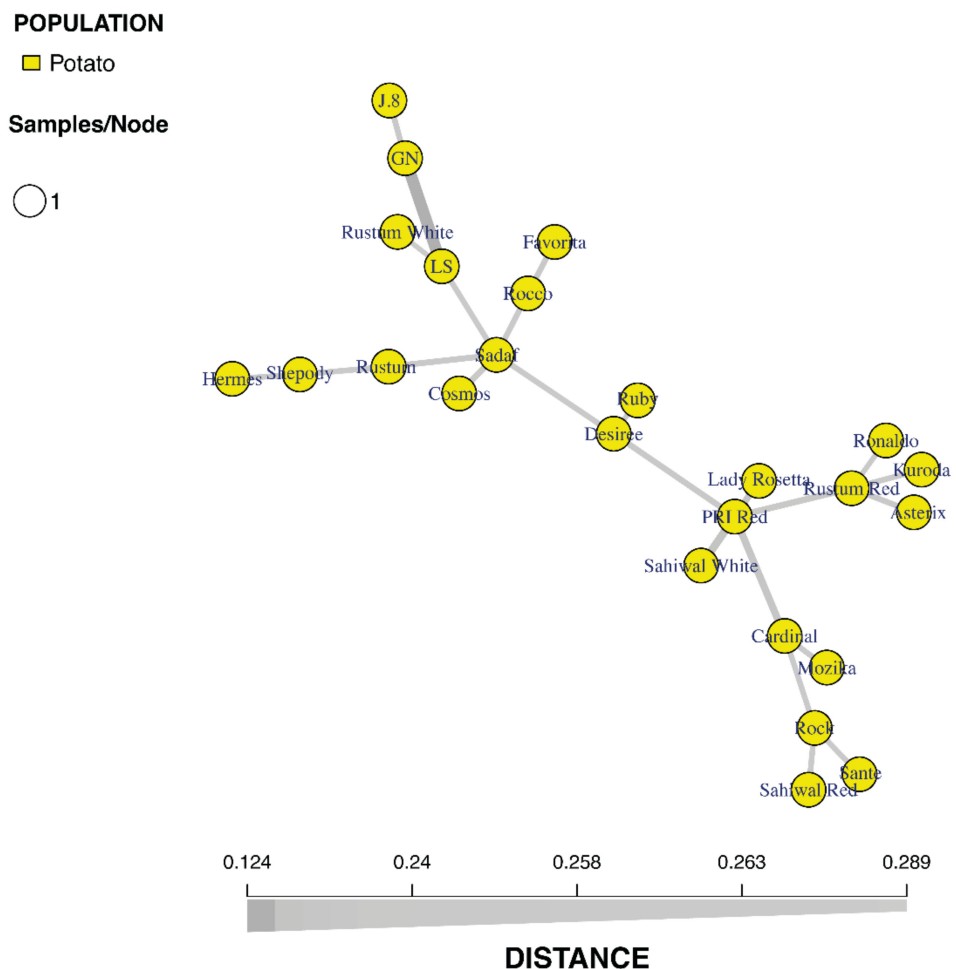

**Figure 2.** Simulation of the Minimum Spanning Network representing 7 populations. The node size and colors represent the number of individuals of each variety of potato. The thickness and color of the ends were proportional to the distance reflecting the number of allelic differences between two individuals.

*3.3. Minimum Spanning Network (MSN) Analysis*

A distinct matrix was used to construct a Minimum Spanning Network (MSN) for the 25 varieties of potato. Each node revealed a different multilogues' variety of potato (MLG) (Figure 2). The node size and colors represent the number of individuals of each variety of potato. The thickness and color of the ends are proportional to the distance reflecting the number of allelic differences between two individuals. The MSN represents the maximum genetic distance of 0.124–0.270 which was observed between the Mozika, Rock, Sahiwal Red, Sante, and J.8 varieties. The PRI Red variety may be the common ancestor of the Lady Rosetta, Rustum Red, Asterix, Ronaldo, Kuroda, and Sahiwal White varieties. Desiree may be the common ancestor of Ruby based on 35 SSR-based markers. Sadaf may be the common ancestor of Cosmos, Rustum, Shepody, Hermes, LS, GN, J.8, Rocco, Rustum White, and Favorita, based on the 35 SSR markers used in our study.

*3.4. Varietal Identification Based on Unique Alleles Based on 35 SSR Markers*

In our current study, 25 varieties were assessed, and 16 potato varieties were uniquely identified by unique markers for potato varieties under the current investigation across the SSR analysis (Table 2). A total of 30 polymorphic SSR bands were found to be variety-specific markers based on their given different sizes. These markers were scored by unique bands for a given variety based on whether they were absent or present. The highest number of variety-specific markers was recorded for Rocco (6 markers), followed by

SSR43016 (130bp), S188 (190bp), STM0031 (250bp), STM1052 (250bp), SSR8242 (280bp), and S25 (700bp), and the lowest number of markers was recorded for the Kuroda 4026/4027 (800bp), S148 (480bp), Rustum Red S25 (185bp), Desiree S192 (185bp), Rock S189 (220bp), Hermes 8242 (480bp), Shepody S170 (90bp), Rustum S187 (200bp), Sadaf 8242 (410bp), and GN S25 (395bp), with only one different marker for each variety. The rest of the potato varieties did not show any specific unique band. The current study indicated that the SSR technique provided suitable differences between all the potato varieties under investigation. (Figures 3 and 4).

**Table 2.** List of 16 Potato varieties that revealed distinctive alleles with specific SSR markers.

| S. No. | Variety Name | SSR Marker |
|---|---|---|
| 1 | Mozika | S174 (310bp), S25 (430bp) |
| 2 | Cardinal | Did not produce a unique allele |
| 3 | Sahiwal White | Did not produce a unique allele |
| 4 | PRI Red | Did not produce a unique allele |
| 5 | Lady Rosetta | Did not produce a unique allele |
| 6 | Kuroda | 4026/4027 (800bp), S148 (480bp) |
| 7 | Rustum Red | S25 (185bp) |
| 8 | Ronaldo | S25 (510bp), S188 (400bp), 4026/4027 (400bp) |
| 9 | Desiree | S192 (185bp) |
| 10 | Rock | S189 (220bp) |
| 11 | Asterix | S174 (230), STM0031 (150bp, 170bp, 250bp) |
| 12 | Ruby | Did not produce a unique allele |
| 13 | Hermes | 8242 (480bp) |
| 14 | Sahiwal Red | Did not produce a unique allele |
| 15 | Sante | Did not produce a unique allele |
| 16 | Shepody | S170 (90bp) |
| 17 | Rustum | S187 (200bp) |
| 18 | Sadaf | 8242 (410bp) |
| 19 | Cosmos | Did not produce a unique allele |
| 20 | Rustum white | 4026/4027 (580bp), ST10032 (130bp) |
| 21 | LS | Did not produce a unique allele |
| 22 | GN | S25 (395bp) |
| 23 | J.8 | Did not produce a unique allele |
| 24 | Favorita | S25 (480bp, 600bp), S148 (550bp) |
| 25 | Rocco | STM1052 (250bp), 43016 (130bp), STM0031 (300bp), 8242 (280bp), S25 (700bp), S188 (190bp) |

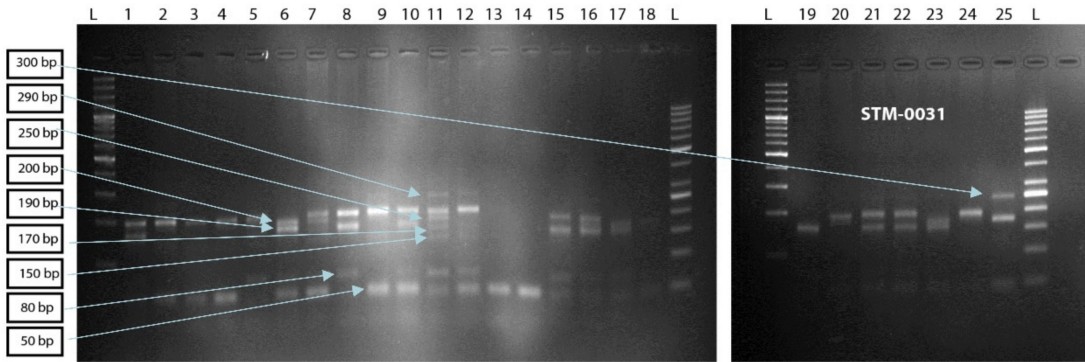

**Figure 3.** The 3% Agarose gel Electropherogram of 25 potato varieties amplified by STM0031 markers. Variety order (1 to 25 from left to right): L = 100bp plus DNA ladder, 1 = Mozika, 2 = Cardinal, 3 = Sahiwal White, 4 = PRI Red, 5 = Lady Rosetta, 6 = Kuroda, 7 = Rustum Red, 8 = Ronaldo, 9 = Desiree, 10 = Rock, 11 = Asterix, 12 = Ruby, 13 = Hermes, 14 = Sahiwal Red, 15 = Sante, 16 = Shepody, 17 = Rustum, 18 = Sadaf, 19 = Cosmos, 20 = Rustum white, 21 = LS, 22 = GN, 23 = J.8, 24 = Favorita, 25 = Rocco, and L = 50bp plus DNA ladder. Blue arrows indicate uniquely identified polymorphic bands in 25 potato varieties. SSR marker 4STM0031 uniquely identified Asterix (STM0031 150bp, 170bp, and 250bp) and Rockco with band sizes of 300bp.

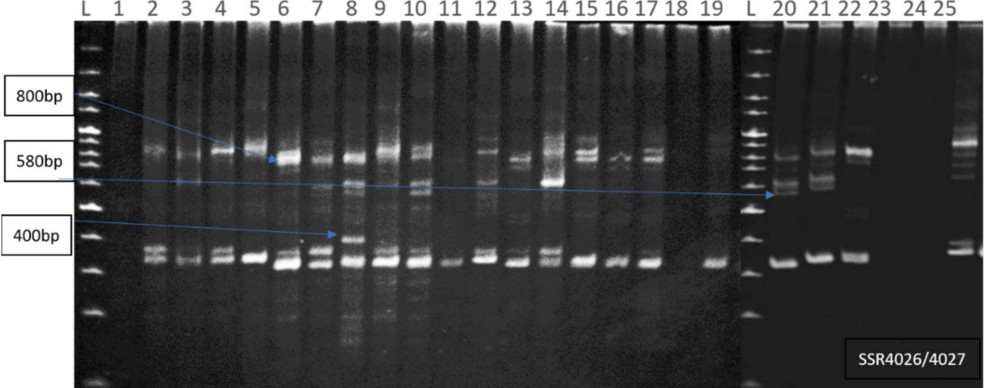

**Figure 4.** The 8% polyacrylamide gel Electropherogram (PAGE) of 25 potato varieties amplified by S4026/4027 markers. Variety order (1 to 25 from left to right): L = 100bp plus DNA ladder, 1 = Mozika, 2 = Cardinal, 3 = Sahiwal White, 4 = PRI Red, 5 = Lady Rosetta, 6 = Kuroda, 7 = Rustum Red, 8 = Ronaldo, 9 = Desiree, 10 = Rock, 11 = Asterix, 12 = Ruby, 13 = Hermes, 14 = Sahiwal Red, 15 = Sante, 16 = Shepody, 17 = Rustum, 18 = Sadaf, 19 = Cosmos, 20 = Rustum white, 21 = LS, 22 = GN, 23 = J.8, 24 = Favorita, 25 = Rocco, and L = 50bp plus DNA ladder. Blue arrows indicate uniquely identified polymorphic bands in sixteen potato varieties. S4026/4027 markers uniquely identified Kuroda (800bp), Ronaldo (580bp), and Rustum White (580bp).

## 4. Discussion

The present investigation is the first report to use the diversity and varietal identification methods, based on thirty-five SSR markers, to differentiate the 25 local potato varieties grown in Pakistan. Therefore, the genotypic management, production of seeds, and protection of breeders' rights are a core concern for the diversity and varietal identification of potato varieties. Morphological approaches for cultivars' identification and diversity determination have been confirmed to be limited due to the influence of environmental factors, low reproducibility, and stability, as well as the need for expertise to maintain the mature plants.

To cope with these limitations and accurately classify cultivars, molecular marker-based DNA fingerprinting can be used [21]. Currently, many scientists use DNA-based molecular markers for studies on the origins of cultivars, fingerprinting patterns, genetic

diversity, and varietal identification [22]. SSR markers are the prevailing method for genetic applications, including diversity analysis, mapping, phylogenetic studies, cultivar difference, and, in breeding programs, marker-assisted selection [23]. SSR markers are extremely polymorphic and conserved sequences are present between and within the closely related cultivars [12]. To develop genomic libraries and E-tag databases, scientists have developed and analyzed SSR markers [6].

SSR markers are extensively utilized for DNA fingerprinting [15], parental investigations [16], genotype relocations [13] and genetic diversity analysis [12] in potatoes. According to the current status, many researchers are working on SSR markers' advancement and adaptability across potato varieties to determine their genetic diversity and achieve varietal identification [17].

In the current findings, 35 polymorphic SSR markers have successfully identified 25 potato varieties grown in Pakistan. Our findings reveal that SSR markers can be used to investigate and differentiate the distinctive varieties of potato grown across Pakistan. However, the genetic distance matrix analysis of the 25 potato varieties based on the 35 polymorphic SSR markers showed a broad variation in these varieties.

The reported findings have revealed that a few SSR primers showed a high number of polymorphisms that can be used to differentiate many potato varieties. In this study, a set of 35 SSR markers amplified a total of 2473 scorable bands and successfully differentiated 25 potato varieties grown in Pakistan. This study is in line with the reported findings in [24], which successfully investigated 138 accessions that were grown in the Western Highlands region of Cameroon through 12 SSR pair markers. The authors of [22] also distinguished 482 potato cultivars conserved in Korea via 24 SSR markers.

In the current study, the maximum number of alleles obtained with the 35 SSR markers is 20 markers higher than the number of alleles obtained from in the previous studies. A total of 4.6 alleles revealed 10 SSR markers that differentiated 38 cultivars in Brazil (Favoretto, P.; et al., 2011) [25]; 10.72 alleles amplified by 25 SSR markers in 482 potato accessions were determined for Chinese cultivars [22]. The maximum number of alleles was observed at 5.12 in 69 German genotypes based on 26 SSR Markers [26]. The maximum number of alleles—amplified by 17 SSR markers—was 5.67 in 14 Northwestern European cultivars [27]. However, in these studies, the results show that the number of alleles was lower than the 22 alleles obtained by using 138 SSR markers to investigate 620 potato cultivars of a Chinese origin [6]. Our study shows a percentage of PIC value that is higher (0.87) than the PIC value of 0.54 revealed in the previous study by [28] and is quite similar to the PIC value of 0.86 determined. The gene diversity based on SSR primers was recorded with an average of (0.87), which is more than and similar to some of the previous findings [29]. The genetic background of any plant material is fully dependent on the genetic diversity parameters such as the nature of primers under study and the number of cultivars. In this study, the observed values in Pakistan are reveal the genetic diversity based on allele number, PIC, and genetic diversity.

In the present findings, the PIC values (0–0.87) and the number of alleles per locus (1–20) are similar to the values reported by [6] with respect to the number of alleles per locus (7–25) and PIC values (0.64–0.93); however, the minimum difference could be attained due to the nature of the primers and genetic background of the genotypes [30]. This experiment was performed on 217 accessions in eight taxonomic groups of cultivated potatoes collected from different origins. Similar studies were also conducted by [30], who differentiated 72 cultivars attained from Japan and the United States through eight sets of SSR primers and detected the number of alleles ranging from 6 to 12, a number of profiles ranging from 16 to 36, and PIC values ranged from 0.72 to 0.94. The authors of [22] also developed 24 SSR markers that revealed a total of 257 alleles, with a mean of 10.72 alleles per locus, which successfully differentiated 482 accessions in Korea. Whereas our study was performed on 25 potato varieties that were differentiated based on thirty-five SSR markers into three major clusters collected from the same origin. The authors of [28] also attained similar findings based on 24 polymorphic SSR markers amplified with a total of 304 bands, of which

297 were found to be polymorphic. In our study, 35 SSR primers were amplified with 2473 bands, of which 196 were polymorphic, sourced from 25 varieties and split into three major clusters that have genetic variations at the molecular level.

From the analysis results in the current study, the Nei's coefficient was found in 25 potato varieties to vary from 0.53–0.95. The Mozika variety is the most diverged compared to all the other potato varieties studied. The maximum genetic distance (0.950) was calculated between Mozika and Sahiwal Red. A high genetic similarity was also shown in Mozika and Ruby at the molecular level. Thus, it could be a useful variety due to its better potential for adaptability against the changing environment. It may be concluded that the cultivars are ecologically very different and belong to far too disparate genetic backgrounds. However, concerning the previous studies, similar results were found, showing a significant correlation between the genetic and the geographic distances that varies from (0.04–0.234). However, the dissimilarity values were found to be high compared to the values obtained in other potato germplasms developed on SSR markers of Ethiopian pepper populations, which observed a minimum distance of 0.48 and a maximum distance of 0.61 [31]. These findings showed low genetic differentiation (0.63 to 0.99) among the 620 cultivars studied by [6]. These findings confirmed differentiation among distinct potato cultivars with a high gene flow.

Unique alleles are fruitfully used for the varietal identification of local potato varieties grown in Pakistan. The maximum number of unique alleles found in the 25 locally grown potato varieties showed that they can be used in potato-breeding programs if they contain phenotypic traits. The present investigation confirmed that the 25 local potato varieties could be classified through SSR primers. In previous research, we employed variety identification strategy and successfully identified 16 local banana varieties using 29 polymorphic SSR markers. Our studies are in line with the previously conducted studies [21] that performed the identification of red-flesh loquat varieties using EST-SSR marker-based cultivar identification. A maximum number of six variety-specific markers were recorded for Rocco, namely, SSR43016 (130bp), S188 (190bp), STM0031 (250bp), STM1052 (250bp), SSR8242 (280bp), and S25 (700bp), and the lowest number of markers was recorded for the PRI Red at S148 (480bp), followed by Kuroda 4026/4027 (800bp), Rustum Red S25 (185bp), Desiree S192 (185bp), Rock S189 (220bp), Hermes 8242 (480bp), Shepody S170 (90bp), Rustum S187 (200bp), Sadaf 8242 (410bp), and GN S25 (395bp), with only one different marker for each variety.

The present study used SSR markers to comprehensively analyze the genetic diversity and perform the varietal identification of 25 potato varieties, and the DNA finger printing of imported varieties was performed in order to identify these varieties from the readily available germplasm at the national level. Furthermore, these varieties have diverse genomes and have the characteristics for a high yield under abiotic stress conditions. These imported high-quality germplasms will be sustainable for use in food in Pakistan; furthermore, this imported germplasm can serve as a reference material for parental lines for developing hybrid potatoes in Pakistan. The imported potato cultivars used in our study can be used as an adopted variety used by farmers for gaining high yields or could be directly used as a variety.

## 5. Conclusions

In this study, we have successfully assessed the genetic diversity and performed the varietal identification of 25 locally grown potato varieties. Nei's genetic diversity and minimum spanning tree-depicted Mozika was found to be the most divergent from the rest of the genotypes. The maximum genetic distance (0.950) was calculated between Mozika and Sahiwal Red. The highest genetic similarity was shown in the Mozika and Ruby varieties at the molecular level. In our current study, 25 varieties were assessed, and 16 potato varieties were uniquely identified by unique markers for potato varieties across the SSR analysis. Therefore, it could be a useful variety due to its better potential adaptability against the changing environment. This is the first report in Pakistan on the

use and effectiveness of SSR markers for assessing genetic diversity and the identification of varieties of potato in Pakistan. Selected markers can be useful for breeders to determine the unequivocal diversity and to perform the identification of the related potato genotypes. A total of 25 varieties must be sequenced and an advanced analysis in terms of SNPs, proteomics, and transcriptomics should be used for further evaluation. In addition, a trait-specific study must be carried out to develop the specific yield and other traits in the germplasm improvement.

**Supplementary Materials:** The following supporting information can be downloaded at: https://www.mdpi.com/article/10.3390/su141811561/s1. Supplementary Table S1: List of SSR markers used for molecular characterization and variety Identification of locally grown 25 potato varieties in Pakistan.

**Author Contributions:** A.M. developed the research concept and helped in funding acquisition. S.N. wrote the original draft of the manuscript, research work and data analysis. K.A., A.S., G.M.A., M.Z. assisted in the literature search and revision. M.N. assisted in research work, manuscript writing, data analysis and formatting, I.H. and S.I.u.H. assisted in the literature search. All authors have read and agreed to the published version of the manuscript.

**Funding:** The research was supported by PSDP (Public Sector Development Program) "Commercialization of potato tissue culture technology in Pakistan" Project number 731.

**Institutional Review Board Statement:** Not applicable.

**Informed Consent Statement:** Not applicable.

**Data Availability Statement:** Not applicable.

**Conflicts of Interest:** The authors declare no conflict of interest.

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
