# Peer review of "Simple Sequence Repeats-Based Genetic Characterization and Varietal Identification of Potato Varieties Grown in Pakistan"

_sustainability, doi:10.3390/su141811561_

Round 1

Reviewer 1 Report

The authors have studied a simple Sequence Repeats based Genetic Characterization and Varietal Identification of Potato Varieties grown in Pakistan. The content of the manuscript is new and brings relevant information to the field. The paper is well written and can be accepted after minor revisions as follows.

#1 There are several typo and grammatical mistakes which should be fixed. Eg., several unnecessary commas and capitalization are observed. Please check the manuscript as a whole. Please check carefully your manuscript.

#2 Abstract:  Every abstract should only contain relevant information that summarizes the work; terms like "therefore", “in addition “... are useless. Also adds quantitative information to the abstract only if relevant. I recommend authors use the following reference to adjust their abstract (https://doi.org/10.1016/j.carbon.2007.07.009).

#3 Introduction: A robustly paragraph about previous works must be added in the introduction section in order to clarify the need to study this topic based on the weak literature. Please divide your introduction into 3 or more paragraphs where we can see some important elements such as background, novelty, justificative, previous works literature, and aim.

#4 Introduction: The novelty of the work is not expressed explicate. In which aspect this work is original and better than others?

#5 Materials and Methods: Please provide the complete list of chemical and reagents with purity in material and method section (materials section). These informations are disperse or missing.

#6 Please move the “Table 2.1” for the support information.

#7 The tables are incorrectly named. The tables must be labeled Table 1, Table 2, ….

#8 The figures are incorrectly named. The tables must be labeled figure 1, figure 2, ….

#9 The acronyms must be defined before their citations. Please check it.

#10 Please provide the complete list of chemical and reagents with purity in material and method section.

#11 References: Please replace old references by latest ones. At least 75% of the references of a modern manuscript should be between 2017-2022. Check carefully your references.

Reviewer 2 Report

The work is interesting. I have a few comments: 1. Do the results obtained by the authors lead only to conclusions about genetic origin? In my opinion, there was no application research factor. The authors should present the results correlated with the yields obtained from 1 acre (data from the ministry). 2. Work on genetics in agricultural areas should be aimed at improving cultivation. What are the factors moving the authors towards improving the potato crop in Pakistan? The results of origin alone are not enough.

Author Response

Author’s Reply to Reviewer 2:

Thank you very much for your valuable comments to improve the quality of the paper. We tried to address all the comments and all the authors confirm that the information has been verified to be correct.

Comment No 1: The work is interesting. I have a few comments: 1. Do the results obtained by the authors lead only to conclusions about genetic origin? In my opinion, there was no application research factor. The authors should present the results correlated with the yields obtained from 1 acre (data from the ministry). 2. Work on genetics in agricultural areas should be aimed at improving cultivation. What are the factors moving the authors towards improving the potato crop in Pakistan? The results of origin alone are not enough.

Authors’ Reply:

  1. i) Do the results obtained by the authors lead only to conclusions about genetic origin?

Yes the main objective of the present study is related to find out genetic diversity, origin and varietal identification that has not been reported before in Pakistan. So, to clear this confusion we differentiated 25 cultivars at molecular level through reported 35 SSR markers that will be used for crossing breeding to improve the yield and quality of the developed varities.

  1. ii) In my opinion, there was no application research factor. The authors should present the results correlated with the yields obtained from 1 acre (data from the ministry).

In the present study there has not been included applied field research factor, because these are mostly established commercial cultivated varieties at Pakistan. We have extracted DNA from 25 varieties of potato and then performed PCR to find out unique novel band. In this regard, field data neither needed nor we sown under field condition for yield purposes. ANOVA (Factorial Design) is applied on morphological data under field grown potato cultivars for comparison. Here, we clearly mentioned molecular data (PCR, SSR) for identification and they do not need factorial design. Under normal research trails i-e., NUYT and DUS are required for yield and uniformity of new developed lines. But our research focus was totally different from the field studies, at this stage we have not developed new cultivars of potato in which yield related data is required, however in future when these will be used for breeding purposes and developed lines will be checked for yield performance. We have only performed molecular analysis to identify existing cultivars already developed by different countries/origin and in Pakistan.  We were focused  on the identification of the cultivars, whether these cultivars are the same or any mixing or mislabeling.

  1. Work on genetics in agricultural areas should be aimed at improving cultivation. What are the factors moving the authors towards improving the potato crop in Pakistan? The results of origin alone are not enough.

As you know the major task of a plant breeder is to develop new cultivars that has the potential of high yield etc. Potato breeders at different countries have domesticated and developed potato cultivars and they have later on distributed their cultivars across the globe. In this regard, Pakistan has obtained different potato cultivars for adoptability and commercial cultivation. There is lack of strong potato breeding programme for the development of new varieties due to many environmental factors and lack of policy interest by the competent authorities. We have goal to identify exotic cultivars through DNA fingerprinting techniques to be used for potato crop improvement in Pakistan.

In other crops, Pakistan is contributing to develop new high yield cultivars, but in case of potato, we need to identify true breeder cultivars (exotic) first and then develop new combinations or cross this material. That’s why we developed this project to test 25 potato varieties based on SSR markers, and found 16 varieties unique and distinct from rest of cultivars included in this study. Therefore focus of this study was not only to find out the origin but distinctness of the varieties to be used in future for biotic and abiotic stress tolerant varieties development in Pakistan with high yield and quality characteristics.

Reviewer 3 Report

 This paper explores the genetic relationship and genetic diversity and provides the basic data for genetic diversity studies and breeding of potato germplasm. After going through the paper, I found some concerns, as listed below:

1.      The introduction section lacks from several important issues: a) The problem definition is unknown, b) the literature review does not follow a straight path, it seems that the authors bring only some references to fulfill the expectations of the readers but it does not conclude to anything important, c) it is not clear what contributions are proposed by the authors.

2.      Most of the ideas written were already described in many literatures. The Authors tried to compile it but lack of the enhancement of the interrelation analysis between the references. It is advised that the authors give a deeper analysis on how these ideas become more applicative strategies so that they can contribute to the next step of implementation.

3.      Avoid lumping references such as [5-7] and others. Instead summarize the main contribution of each referenced paper in a separate sentence and by including the reference number.

4.      The novelty of the present work should be well stated and justified. The new author's contribution should be justified regarding the previous works in the literature. The literature review should be updated to help readers better understand the subject matter and novelty aspects of this work compared to the recently published works.

5.      More in-depth analysis of the author's contribution of this paper in the introduction section.

6.      In my opinion, the model construction is simple; moreover, there are several up-to-date approaches for the idea. Authors should look for these approaches, compare the results and prove their idea. This is the major concern.

7.      Figures 3-3, 3-4 need to be enhanced.

Author Response

Response to Reviewer 3 Comments

Thank you very much for your valuable comments to improve the quality of the paper. We tried to address all the comments and all the authors confirm that the information has been verified to be correct.

The necessary changes have been made to the paper.

Comment No 1:

The introduction section lacks from several important issues: a) The problem definition is unknown, b) the literature review does not follow a straight path, it seems that the authors bring only some references to fulfill the expectations of the readers but it does not conclude to anything important, c) it is not clear what contributions are proposed by the authors.

Authors’ Reply: Paragraph 1, 2 and 3 are added according to suggestions in introduction section and divided accordingly into 5 paragraphs. Highlighted portion incorporated Page 2 and 3. Changes are made to clarify the confusion.

Comment No 2:

Most of the ideas written were already described in many literatures. The Authors tried to compile it but lack of the enhancement of the interrelation analysis between the references. It is advised that the authors give a deeper analysis on how these ideas become more applicative strategies so that they can contribute to the next step of implementation.

Authors’ Reply:

Changes are made according to suggestion. Paragraph added in introduction section which explains the importance of this research work compared with previous literature. Latest references have been incorporated in introduction section.

Comment No 3:

Avoid lumping references such as [5-7] and others. Instead summarize the main contribution of each referenced paper in a separate sentence and by including the reference number.

Authors’ Reply:

Changes are made according to suggestions. Latest references have been incorporated in whole manuscript.

Comment No 4:

The novelty of the present work should be well stated and justified. The new author's contribution should be justified regarding the previous works in the literature. The literature review should be updated to help readers better understand the subject matter and novelty aspects of this work compared to the recently published works.

Authors’ Reply:

Changes are made according to suggestions. Line ….

Comment No 5:

The novelty of the present work should be well stated and justified. The new author's contribution should be justified regarding the previous works in the literature. The literature review should be updated to help readers better understand the subject matter and novelty aspects of this work compared to the recently published works.

Authors’ Reply:

Yes, your comment is valid. Added paragraph in introduction section by highlighting the novelty of this manuscript. Page 2 and 3 which represents the novelty of this manuscript.

Comment No 6:

More in-depth analysis of the author's contribution of this paper in the introduction section.

Authors’ Reply:

Changes are made according to the suggestions. Latest references have been incorporated in introduction section. Highlighted in introduction and discussion section.

Comment No 6:

In my opinion, the model construction is simple; moreover, there are several up-to-date approaches for the idea. Authors should look for these approaches, compare the results and prove their idea. This is the major concern.

Authors’ Reply:

Changes are made according to suggestions compare with the updated approaches.

Comment No 7:

Figures 3-3, 3-4 need to be enhanced.

Authors’ Reply:

Changes are made according to suggestions. Figure 3 (Page 8), Figure 4 (Page 9).

Round 2

Reviewer 2 Report

Accept in present form

Author Response

Thank you for your comments

Reviewer 3 Report

Authors addressed my comments

Author Response

Thank you for your comments